# Base-Free Oxidation of HMF to FDCA over Ru/Cu-Co-O·MgO under Aqueous Conditions

**DOI:** 10.3390/molecules29133213

**Published:** 2024-07-06

**Authors:** Shuang Zhang, Guoning Chu, Sai Wang, Ji Ma, Chengqian Wang

**Affiliations:** Institute of Petrochemical Technology, Jilin Institute of Chemical Technology, 45 Chengde Street, Jilin 132022, China; chu2269566@126.com (G.C.); ws18435221867@126.com (S.W.); m15682902278@126.com (J.M.); wangchengqian1992@126.com (C.W.)

**Keywords:** ruthenium (Ru), 5-hydroxymethylfurfural (HMF), base-free oxidation, 2,5-furandicarboxylic acid (FDCA), product separation

## Abstract

The copper–cobalt metal oxide composite magnesium oxide catalyst loaded with Ru has achieved the aerobic oxidation of 5-hydroxymethylfurfural (HMF) to the bio-based polyester monomer 2,5-furandicarboxylic acid (FDCA) under base-free conditions. Several Ru/Cu-Co-O·MgO catalysts were prepared, with Cu-Co-O being a combination of CuO and Co_3_O_4_. The catalyst’s activity was boosted by the synergistic interaction between copper and cobalt, as well as an optimal copper-to-cobalt molar ratio. Optimal catalytic activity was observed in the Ru_4_/Cu1-Co1-O·MgO catalyst, loaded with 4 wt% Ru when copper-to-cobalt molar ratio of 1:1 and magnesium oxide compounding amount of 6 mmol were employed. The inclusion of MgO and the load of Ru not only expanded the specific surface area of the catalyst but also heightened its basicity. Additionally, the presence of loaded Ru improved the catalyst’s reducibility at low temperatures. In aqueous solution under oxygen pressure, the conversion rate of HMF achieved 100%, and the yield of FDCA was 86.1%. After five reaction cycles, examining the catalyst and solution revealed that Ru nanoparticles resisted leaching or oxidation, and MgO exhibited only slight dissolution. The green separation of the product was achieved using semi-preparative liquid chromatography, selectively collecting the FDCA-containing solution by exploiting variations in interactions between solutes and the stationary/mobile phases. The subsequent steps involved rotary evaporation and drying, resulting in FDCA powder with a purity exceeding 99%. Notably, this approach eliminated the need to introduce concentrated hydrochloric acid into the system for FDCA separation, providing a novel method for synthesising powdered FDCA.

## 1. Introduction

Amid growing global interest in renewable resources and green chemistry, chemicals sourced from biomass are gaining prominence, drawing considerable research attention in both academic and industrial spheres [1]. 5-Hydroxymethylfurfural (HMF), a crucial intermediate derived from biomass, plays a key role in transforming biomass into valuable chemicals. Recently, interest in the further conversion of HMF into the bio-based monomer 2,5-furandicarboxylic acid (FDCA) is increasing among researchers.

Polyethylene furanoate (PEF), a biodegradable polymer synthesised using FDCA as the monomer, presents environmental advantages compared to petroleum-based polyethylene terephthalate (PET) [2]. Furthermore, compared to PET, PEF demonstrates enhanced physical properties, including superior thermoplasticity and gas barrier characteristics [3]. PEF plastics, known for their resistance to deformation at high temperatures, are suitable for producing packaging bags, bottles, and containers. Processing recycled PEF into fibres can also be utilised to manufacture 100% biobased T-shirts, contributing to further reduction in carbon emissions.

In the oxidative conversion of HMF to FDCA, precious metal catalysts play a crucial role due to their remarkable ability to activate molecular oxygen. Notably, Pt [4], Au [5], and Pd [6] have demonstrated exceptional catalytic performance in this process. Nevertheless, their industrial application faces challenges due to their high economic cost or the requirement to add liquid alkali. Consequently, developing a cost-effective catalyst capable of oxidising HMF to FDCA under base-free conditions is important.

In contrast to the previously indicated precious metals, Ru is comparatively cost-effective and displays a broader valence state range (from −2 to +8). Ru can be supported on diverse carriers. Our research group developed an Ru_4_/Cr2-Fe1-O catalyst and employed an aqueous solution of KHCO_3_ as the solvent. Operating under an oxygen pressure of 1 MPa at 100 °C for 16 h, the conversion rate of HMF was 100%, and the yield of FDCA was 99.9%. This catalyst displayed robust stability, retaining high catalytic activity after five cycles [7]. However, the catalytic reaction necessitated liquid alkali conditions. In addressing this limitation, Antonyraj et al. introduced an Ru/MgAlO catalyst that can oxidize HMF to FDCA without the need for a base additive [8]. The reaction proceeded at 140 °C and 90 psi oxygen pressure for 4 h, with complete HMF conversion and 99% FDCA yield. While the catalytic performance was satisfactory, there was an issue with the leaching of the active component Ru, resulting in reduced stability. The Ru/HAP catalyst reported by Gao et al. was used in a base-free aqueous solution under an oxygen pressure of 1 MPa and 120 °C for 24 h; under these conditions, HMF can be completely converted, resulting in an FDCA yield of 99.6% [9]. However, it is worth noting that the oxidation of Ru^0^ was identified as a significant factor in catalyst deactivation, evidenced by a 12% reduction in the proportion of Ru^0^ after the reaction.

MgO, as an alkaline oxide, can replace the addition of liquid alkali. Gorbanev et al. produced an Ru(OH)_x_/MgO catalyst, demonstrating its efficacy in the oxidation of HMF to FDCA without the need for alkaline conditions [10]. Completing the reaction at 2.5 bar oxygen pressure and 140 °C for 20 h resulted in a conversion rate of 100% HMF and a 100% yield of FDCA. However, the catalyst underwent reaction with the product FDCA during the process, leading to the leaching of Mg^2+^. ICP analysis of the solution after the reaction revealed that the concentration of Mg^2+^ in the solution reached 1.59 g/L.

Furthermore, the characteristics of the supports significantly impact the performance of catalysts. On the one hand, the interaction between Co and another metal oxide can influence redox properties and generate diverse oxygen species on the catalyst [11]. On the other hand, Cu exhibited strong oxygen affinity and, when combined with an active metal [12,13], serves as an effective promoter. Additionally, Cu can act as structural promoter, preventing the agglomeration of the active metal. The synergistic effect between Cu and Co effectively facilitates electron transfer, thereby enhancing the yield of FDCA.

Consequently, copper–cobalt bimetallic oxide composite with MgO as the support, further loaded with Ru as the catalyst (Ru/Cu-Co-O·MgO), was prepared. This catalyst can oxidize HMF to FDCA in aqueous solution under base-free conditions, with the advantage that Ru nanoparticles are resistant to leaching and oxidation, and the leaching of Mg particles is mitigated. First, incorporating MgO and loading Ru effectively enhanced the catalyst’s alkaline sites, facilitating the catalysis of HMF oxidation to FDCA under base-free conditions. Secondly, the improved low temperatures reducibility and the synergistic interaction between copper and cobalt enhanced the catalyst’s activity. Under optimal reaction conditions, Ru_4_/Cu1-Co1-O·MgO loaded with 4 wt% Ru achieved a 100% HMF conversion rate and a maximum FDCA yield of 86.1%. The FDCA solution was separated from the reaction mixture using semi-preparative liquid chromatography, followed by rotary evaporation and drying. This process yielded powdered FDCA with a purity exceeding 99%. Notably, this separation method is more environmentally friendly and convenient than existing separation methods, eliminating the need to add concentrated hydrochloric acid.

## 2. Results and Discussion

### 2.1. Structure of Catalysts

Figure 1 displays the XRD spectra of Cu-Co-O, Cu-Co-O·MgO, Cu_x_O_y_, Co_x_O_y_, Cu_x_O_y_·MgO, Co_x_O_y_·MgO, and MgO. In Figure 1a, it is evident that the peaks observed for Cu_x_O_y_ at 32.5°, 35.5°, 38.7°, and 48.7° correspond to the monoclinic structure of tenorite CuO (JCPDS#45-0937) on the (−110), (002), (111), and (−202) planes, respectively [14]. Regarding Co_x_O_y_, the peaks observed at 19.0°, 31.2°, 38.8°, 44.8°, 59.3°, and 65.2° were consistent with the face-cantered cubic structure of Co_3_O_4_ (JCPDS#42-1467) on the (111), (220), (311), (400), (511), and (440) planes [15]. Furthermore, with the increase in the copper–cobalt molar ratio, the diffraction peak intensity of CuO in the Cu-Co-O samples was enhanced, and these sharper peaks appeared without notable shifts in their positions. These XRD findings confirmed that Cu-Co-O was a composite material of CuO and Co_3_O_4_.

As shown in Figure 1b, in addition to the diffraction peaks for CuO and Co_3_O_4_, a set of new diffraction peaks emerged at 37.1°, 43.0°, 62.4°, 74.8°, and 78.7° in the XRD spectra of a series of catalysts. These peaks aligned with the (111), (200), (220), (311), and (222) planes of the polycrystalline cubic structure of MgO (JCPDS#87-0653), respectively [16]. This observation indicated the successful incorporation of MgO into Cu-Co-O, Cu_x_O_y_, and Co_x_O_y_, forming composite metal oxides.

The pore structure and surface area of the prepared samples were analysed through N_2_ adsorption–desorption isotherms, and the corresponding results and parameters were outlined in Figure 2 and Table 1. It was evidenced that the N_2_ adsorption–desorption curves of all samples exhibit Type IV isotherms, with a characteristic H3 type hysteresis loop occurring in the relative pressure (P/P_0_) range of 0.6–1.0. This observation suggested the existence of slit-like pores formed by the aggregation of flake-like particles [17].

Upon compositing with MgO, there was a notable improvement in both the specific surface area and pore volume of the Cu-Co-O samples. Specifically, after Cu1-Co2-O, Cu1-Co1-O, and Cu2-Co1-O samples were composite with MgO, their specific surface areas significantly increased from 1.3 m^2^/g, 4.0 m^2^/g, and 6.6 m^2^/g to 39.3 m^2^/g, 35.9 m^2^/g, and 29.9 m^2^/g, respectively. Additionally, the pore volumes increased from 0.0034 cm³/g, 0.0190 cm³/g, and 0.0268 cm³/g to 0.1264 cm³/g, 0.0677 cm³/g, and 0.1070 cm³/g, respectively. Following the incorporation of MgO, the average pore diameter decreased, from 10.87 nm, 18.84 nm, and 15.76 nm to 2.73 nm, 7.49 nm, and 3.83 nm, respectively. This phenomenon could be due to the formation of additional mesoporous channels characterised by smaller average diameters following the combination with MgO. This modification contributed to the observed increase in the specific surface area.

After loading Ru on Cu-Co-O·MgO, an enhancement was observed in the specific surface area and pore volume. Specifically, the specific surface areas for Ru_4_/Cu1-Co2-O·MgO, Ru_4_/Cu1-Co1-O·MgO, and Ru_4_/Cu2-Co1-O·MgO were 44.4 m^2^/g, 55.9 m^2^/g, and 63.4 m^2^/g, respectively. Correspondingly, the pore volumes increased to 0.2041 cm³/g, 0.2193 cm³/g, and 0.3358 cm³/g, respectively. Furthermore, the average pore diameters of Ru_4_/Cu1-Co2-O·MgO and Ru_4_/Cu2-Co1-O·MgO decreased from 2.73 nm and 3.83 nm to 1.16 nm and 2.45 nm, respectively. The observed increase in the specific surface area and pore volume, coupled with the reduction in average pore diameter for Ru_4_/Cu1-Co2-O·MgO and Ru_4_/Cu2-Co1-O·MgO, could be attributed to the loading of Ru, which contributes to a more abundant mesoporous channel structure. This, in turn, further enhances the specific surface area of the catalyst. Loading Ru to Cu1-Co1-O·MgO resulted in an increased specific surface area and pore volume, with the average pore diameter expanding from 7.49 nm to 15.68 nm. The increase in specific surface area, pore volume, and average pore diameter for Ru_4_/Cu1-Co1-O·MgO probably contributed to the appropriate Cu-Co molar ratio that facilitates improved interactions between Ru and the support. These strengthened interactions may result in the amalgamation of initially smaller pores, contributing to the observed changes in the catalyst’s pore characteristics.

As shown in Table 1, the Ru_4_/Cu2-Co1-O·MgO catalyst exhibits the highest specific surface area (63.4 m^2^/g), followed by Ru_4_/Cu1-Co1-O·MgO (55.9 m^2^/g), with Ru_4_/Cu1-Co2-O·MgO having the lowest (44.4 m^2^/g). Despite Ru_4_/Cu2-Co1-O·MgO having the largest specific surface area among these catalysts, the Ru_4_/Cu1-Co1-O·MgO catalyst demonstrated the highest catalytic activity. This indicates that the specific surface area was not the sole factor influencing catalytic activity.

### 2.2. Evaluation of the Surface Chemistry of Catalysts

The alkalinity of the catalysts was assessed through CO_2_-TPD, and the results were depicted in Figure 3 and summarised in Table 2. The total alkalinity increased for the Cu1-Co1-O support after compositing with MgO or loading with Ru. This suggested that incorporating MgO or introducing Ru led to the generation of basic sites on the catalyst support. In the case of the Cu1-Co1-O support, when it was both composited with MgO and loaded with Ru, the resulting Ru_4_/Cu1-Co1-O·MgO catalyst exhibited more robust basic sites, measuring 0.293 mmol/g. In base-free aerobic oxidation reactions of alcohols, basic sites can accelerate the activation of hydroxyl groups. Typically, a higher total alkalinity can enhance the base-free oxidation of HMF to FDCA [4]. This might be the reason for the high catalytic activity of the Ru_4_/Cu1-Co1-O·MgO catalyst in the oxidation reaction of HMF [8].

The H_2_-TPR profiles of various catalysts were depicted in Figure 4. Notably, in the case of Cu1-Co1-O, the reduction peak around 268 °C can be attributed to the reduction of CuO to Cu. Additionally, the broad peak observed between 286 °C and 344 °C indicated the reduction of Co^3+^ to Co^2+^ and the subsequent reduction of Co^2+^ to Co^0^ [18].

In the case of Cu1-Co1-O·MgO, the reduction peak observed around 268 °C corresponded to the reduction of CuO to Cu. Furthermore, the reduction peaks identified around 313 °C and 360 °C can be attributed to the reduction of Co^3+^ to Co^2+^ and the subsequent reduction of Co^2+^ to Co^0^, respectively [18].

The H_2_-TPR profile of Ru_4_/Cu1-Co1-O clearly showed that after loading Ru, the reduction temperature of the catalyst shifted to lower temperatures compared to the Cu1-Co1-O support. This shift suggested the reduction of Ru species, and partial hydrogen overflowed onto the surface of the support, ultimately leading to the reduction of the support. The initial reduction peak observed at 156 °C indicated the reduction of RuO_x_ to metallic Ru, signifying that the catalyst loaded with Ru demonstrates improved low-temperature reducibility [7]. The broad peak following 221 °C can be attributed to the reduction of CuO to Cu and Co^3+^ to Co^2+^. The reduction peak around 495 °C was the further reduction of Co^2+^ to Co^0^ [19].

In the H_2_-TPR profile of Ru_4_/Cu1-Co1-O·MgO, it was noted that the reduction temperature shifted toward lower temperatures, similar to those observed with the Cu1-Co1-O support. However, compared to Ru_4_/Cu1-Co1-O, the reduction temperature shifted toward higher temperatures after compositing with MgO. The reduction peak observed around 174 °C can be attributed to the reduction of Ru interacting with MgO [20]. This outcome suggested a robust interaction between Ru species and MgO, impeding the further reduction of Ru and resulting in higher reduction temperatures. Moreover, an optimal amount of Ru loading and compositing with MgO prevents the agglomeration of Ru species with MgO [21]. The reduction peak at 218 °C corresponds to the reduction of CuO to Cu, while the peak at 337 °C signifies the reduction of Co^3+^ to Co^2+^. The reduction peak around 461 °C can be attributed to the further reduction of Co^2+^ to Co^0^.

The XPS spectra of the Ru_4_/Cu1-Co1-O·MgO catalyst were further analysed to explore the elemental distribution and valence states of the samples. As shown in Figure 5, the presence of Ru, Cu, Co, O, and Mg elements were evidenced. In Figure 5a, the Mg 1s XPS spectrum displayed a primary peak at 1303.2 eV, indicating Mg in the form of Mg^2+^. Figure 5b presented the XPS spectrum for Cu 2p, featuring a higher binding energy peak at 932.2 eV (Cu 2p_3/2_) and a lower binding energy peak at 951.9 eV (Cu 2p_1/2_). The spin–orbit splitting value between Cu 2p_3/2_ and Cu 2p_1/2_ was 19.7 eV, suggesting the presence of Cu in the form of CuO [22]. A satellite peak observed 9.2 eV ahead of the main peak Cu 2p_3/2_ indicated the presence of Cu in the form of Cu^2+^ [23]. Figure 5c presents the XPS spectrum for Co 2p, featuring two primary peaks at 780.2 eV and 795.9 eV, corresponding to Co 2p_3/2_ and Co 2p_1/2_, respectively. The spin–orbit splitting value of 15.7 eV between Co 2p_3/2_ and Co 2p_1/2_ indicated the coexistence of Co^2+^ and Co^3+^ in the catalyst [24]. Figure 5d displayed the XPS analysis for O 1s, split into three peaks: a peak at 530.4 eV labelled as O_lattice_, accounting for 32.5% of the relative fraction; a peak at 531.1 eV labeled as O_surface_, with a relative fraction of 35.6%; and a peak at 531.8 eV labeled as O_mw_, with a relative fraction of 31.9%. The O_lattice_ peak was assigned to lattice oxygen as “O^2−^”, acting as a nucleophilic agent that reacts with adsorbed hydrogen generated by dehydrogenating hydroxyl groups in HMF, producing water. This process facilitates the oxidation of HMF [25]. The O_surface_ peak was linked to oxygen species adsorbed on the catalyst surface, such as “O^−^” and “O_2_^2−^”, characterised by higher mobility and easy activation. These surface oxygen species contribute to enhanced catalytic activity [26]. The O_mw_ peak was ascribed to adsorbed water molecules. Figure 5e displays the XPS spectrum for Ru 3p, revealing the presence of Ru primarily in the forms of Ru^0^ and Ru^4+^. The peaks at binding energies of 463.1 eV and 485.1 eV correspond to Ru 2p_3/2_ and Ru 2p_1/2_, respectively, with a spin–orbit splitting energy of 22 eV, indicating the presence of Ru^0^ and Ru^4+^ on the material’s surface [27]. Ru^0^, constituting 60.1% of the Ru species on this catalyst, plays a crucial role in facilitating the selective oxidation of HMF to FDCA. In this particular catalyst, Ru^0^ was more abundant compared to Ru^4+^. Ru was the active centre in Ru_4_/Cu1-Co1-O·MgO, with Ru^0^ being the species actively promoting the reaction [9]. In addition, the mass proportion of the Ru element in Ru_4_/Cu1-Co1-O·MgO catalyst was determined to be 3.95 wt% through EDS, which basically conforms to the theoretical value (Appendix A).

### 2.3. Morphological Analysis of the Catalyst

The SEM images of the catalysts were depicted in Figure 6. As shown in Figure 6a, Cu1-Co1-O exhibited block-like structures. Figure 6b revealed that MgO displays a layered flake structure. As indicated in Figure 6c, after Cu1-Co1-O composite with MgO, block-like structures were stacked on the layered flakes of MgO, indicating that MgO was successfully composite with Cu1-Co1-O. Figure 6d showed that the structure of the Ru_4_/Cu1-Co1-O·MgO catalyst was also composed of layered flakes and block-like structures.

Figure 7a shows that Cu1-Co1-O exhibits quasi-spherical morphology. Figure 7b,d contains TEM images of Cu1-Co1-O·MgO. In Figure 7d, the lattice fringe spacing of 0.149 nm corresponds to the interplanar spacing of MgO (220), indicating the successful compositing of MgO with Cu1-Co1-O. Figure 7c,e are TEM images of Ru_4_/Cu1-Co1-O·MgO. Figure 7e showed that the lattice fringe spacing of 0.136 nm corresponds to the interplanar spacing of Ru (110), demonstrating the successful loading of Ru onto the Cu1-Co1-O·MgO catalyst.

### 2.4. Optimisation of Catalyst Preparation Conditions

The oxidative synthesis of FDCA from HMF was explored to identify the optimal support by manipulating the copper-to-cobalt molar ratios, the amount of magnesium oxide composite, and the ruthenium loading.

Table 3 (entries 1–6) revealed that Ru_4_/MgO attained a 59.3% FDCA yield. After compounding with Cu1-Co1-O, the FDCA yield of the Ru_4_/Cu1-Co1-O·MgO catalyst increased to 78.6%. This indicates a synergistic interaction between MgO and the copper–cobalt bimetallic oxides. By varying the Cu:Co molar ratios (1:0, 1:1, 1:2, 2:1, 0:1), it was found that the highest catalyst activity occurred at a Cu:Co molar ratio of 1:1. This identified the optimal copper-to-cobalt ratio for the support as 1:1. Notably, the activity of Ru_4_/Cu_x_O_y_·MgO and Ru_4_/Co_x_O_y_·MgO catalysts was lower than that of the Ru_4_/Cu1-Co1-O·MgO catalyst, emphasising a synergistic effect between Cu and Co; an appropriate balanced Cu-Co molar ratio proves advantageous for augmenting the catalyst’s activity.

The impact of the amount of composited MgO on catalytic activity was explored, as illustrated in Table 3 (entries 6–10). In the absence of the MgO composite, HMF achieved complete conversion, yet the FDCA yield remained relatively low at 10.6%, possibly due to the catalyst’s lack of sufficient basic sites. As the amount of composited MgO increased, the yield of FDCA also increased. With the amount of composited MgO being 6 mmol, the FDCA yield peaked at 78.6%. However, an increase in the amount of composited MgO resulted in a decline in FDCA yield. This reduction can be ascribed to the increased amount of composited MgO, which reduces the content of CuCo in the Ru_4_/Cu-Co-O·MgO catalyst, consequently reducing the catalyst’s activity. This implies that maintaining a suitable proportion of MgO proves advantageous in boosting the activity of the Ru_4_/Cu-Co-O·MgO catalyst. Consequently, the ideal amount of composited MgO was identified as 6 mmol. These findings proposed that the alkaline sites from MgO play a role in facilitating the oxidation of HMF to FDCA in the absence of a base.

The impact of Ru loading on catalyst activity was examined, as outlined in Table 3 (entries 11–15). Without Ru loading, the Cu1-Co1-O·MgO catalyst exhibited an HMF conversion rate of 72.7%, with a mere 3.5% FDCA yield. Upon loading 1 wt% of Ru, the HMF conversion rate and FDCA yield experienced an increase. Upon reaching 4 wt% Ru loading, a complete conversion of HMF occurred, yielding the highest FDCA output of 78.6%. Subsequent increases in Ru loading resulted in a decline in the FDCA yield, potentially attributable to catalyst particle agglomeration induced by excessive Ru loading [28], leading to reduced catalyst activity. Consequently, the optimal Ru loading amount was identified as 4 wt%. These observations suggest that Ru serves as the active centre in this catalyst.

### 2.5. Evaluation of Catalyst Activity

To determine the most effective reaction conditions for the oxidative synthesis of FDCA from 0.2 mmol HMF in a 5 mL water solution employing the Ru_4_/Cu1-Co1-O·MgO catalyst, and exploration into the influences of reaction temperature, reaction time, catalyst quantity, and reaction pressure on catalytic activity was conducted.

Figure 8a illustrates the impact of reaction temperature on catalyst activity. At 90 °C, HMF conversion approached completeness, yet the FDCA yield remained relatively low at 27.3%. With increased reaction temperature, the FDCA yield increased, peaking at 86.1% at 120 °C. However, further temperature increases led to a decline in the FDCA yield, possibly attributed to enhanced temperatures promoting additional reactions of FDCA, resulting in the formation of humins or other oxidation products through ring-opening reactions [29]. Consequently, the optimal reaction temperature was identified as 120 °C. TON and TOF values are crucial for assessing catalyst performance. TON and TOF values were calculated by evaluating the Ru_4_/Cu1-Co1-O·MgO catalyst across various reaction temperatures. As indicated in Table 4, there is a discernible trend; with an increase in reaction temperature, both TON and TOF continuously increased. The peak of TON and TOF values was attained at a reaction temperature of 120 °C. However, a substantial decrease in TON and TOF values was observed upon further increasing the temperature to 130 °C. This decrease could be ascribed to increased by-product formation and the intensified reaction of FDCA, resulting in the generation of humins as the temperature increased.

Figure 8b shows that the catalyst activity is affected by the reaction time. After 2 h of reaction, the HMF conversion rate reached 58.1%, yielding 5.3% FDCA, with HFCA and FFCA yields at 11.4% and 15.7%, respectively. With prolonged reaction time, the HMF conversion rate and FDCA yield increased. After 8 h, HMF achieved complete conversion, yielding 62% FDCA, with HFCA and FFCA yields at 5.8% and 11.4%, respectively. At 12 h, the FDCA yield peaked at 86.1%, accompanied by HFCA and FFCA yields of 1.2% and 1.8%, respectively. However, extending the reaction time beyond this point reduced the FDCA yield, potentially attributing to excessively prolonged reaction times leading to side reactions such as ring-opening and the polymerisation of FDCA [30]. Based on this analysis, the optimal reaction time was established at 12 h.

Figure 8c depicts the effect of the catalyst amount on catalytic activity. Adjustments in the amount of catalyst directly impact the number of active sites on the catalyst, consequently influencing the reaction activity. Upon introducing 0.02 g of catalyst into the reaction system, the HMF conversion rate reached 91.9%, with an FDCA yield of 33.5%. A substantial amount of by-products was evident, with HFCA and FFCA yields reaching 16.5% and 35.4%, respectively. With an incremental increase in the amount of catalyst in the reaction system, there was a decline in by-product yield and a simultaneous increase in the FDCA yield. Upon reaching 0.08 g of catalyst, complete conversion of HMF occurred, yielding 86% FDCA. Continued increments in the amount of catalyst resulted in a reduction in FDCA yield, potentially attributed to the overoxidation of HMF, leading to the formation of other by-products [31]. Consequently, 0.08 g of catalyst was chosen for subsequent studies.

Figure 8d demonstrates how varied oxygen pressures influence catalyst activity. O_2,_ chosen for its environmentally friendly nature and mild oxidative activity, was the oxidising agent for the HMF oxidation reaction. At an oxygen pressure of 0 MPa, HMF achieved near-complete conversion, with a conversion rate of 99.1% and an FDCA yield of only 9.6%. The impact of oxygen pressure on the FDCA yield was notable, exhibiting a pronounced increase in the FDCA yield with increasing oxygen pressure. At an oxygen pressure of 1 MPa, HMF achieved complete conversion, resulting in an FDCA yield of 86.1%. These experimental findings emphasise that adequate oxygen pressure can effectively promote progress of the catalytic reaction.

The stability and recyclability of catalysts are crucial considerations for their practical application and potential industrialisation. Cyclic experiments were undertaken using the Ru_4_/Cu1-Co1-O·MgO for catalytic reaction under optimal conditions. The outcomes, depicted in Figure 9, reveal that the FDCA yield declined from 86.1% to 75.4% after five cycles.

To confirm the cause of catalyst deactivation, XPS analysis was carried out on the catalyst. An ICP-OES test was conducted on the solution post-reaction.

Figure 10 presents the analysis of the catalyst after five reaction cycles through XPS testing. As depicted in Figure 10a, following five cycles, the proportion of O_lattice_ increased from 32.4% to 42.5%, the O_surface_ rose from 35.7% to 46.3%, and the O_mw_ decreased from 31.9% to 11.2%. In Figure 10b, it was evidenced that Ru^0^ underwent partial oxidation to Ru^4+^ during the cycling process, with its proportion decreasing from 60.1% to 58.6% after five reaction cycles.

The analysis of the reaction solution after five cycles was conducted using ICP-OES to investigate the potential dissolution of MgO and the leaching of Ru nanoparticles during the reaction. According to Table 5, Ru nanoparticles in the Ru_4_/Cu1-Co1-O·MgO catalyst are not prone to leaching. Compared to the Ru_4_/MgO catalyst, the dissolution of MgO was inhibited, likely due to the synergistic interaction between the copper–cobalt bimetallic oxides and MgO [32].

### 2.6. Potential Reaction Mechanism of the Catalyst

Considering the experimental results outlined earlier, the potential reaction mechanism of the Ru_4_/Cu1-Co1-O·MgO catalyst was discussed, as shown in Figure 11. In this proposed reaction pathway, the aldehyde side of HMF is adsorbed on the Ru^0^ active sites, initiating the hydrolysis reaction that yields an intermediate diol. Concurrently, the oxygen adsorbed on the catalyst surface is activated, converting to active lattice oxygen at the oxygen vacancy sites of the catalyst support. On the Ru^0^ active sites, the O-H bond of the intermediate diol undergoes cleavage, with the two liberated hydrogen atoms combining with the active lattice oxygen on the catalyst surface to generate water. This process completes the dehydrogenation reaction of the intermediate diol, leading to the production of HFCA [33]. The formed HFCA is promoted by the surface alkaline centre provided by MgO [34], and its hydroxyl group adsorbed on the catalyst surface. Through the cleavage of the O-H bond, one hydrogen atom is removed, resulting in the formation of a metal–alkoxide intermediate. Subsequently, β-H elimination reaction removes the H atom connected to the carbon. This oxidation process converts the hydroxyl group into the aldehyde group, generating the second intermediate, FFCA [35]. Throughout this process, electrons are transferred to the catalyst surface. In the concluding steps, the aldehyde side chain of FFCA engages in the hydrolysis reaction at the Ru^0^ active sites, leading to the regeneration of the intermediate diol. This diol undergoes a subsequent dehydrogenation reaction, removing two hydrogen atoms. These liberated hydrogen atoms then combine with the active lattice oxygen on the catalyst surface, forming water and ultimately resulting in the desired product, FDCA. Based on the intermediates produced, the hypothesised reaction pathway of the catalyst can be summarised as HMF → HFCA → FFCA → FDCA. The reduction of Co^3+^ to Co^2+^ consumes surplus electrons on the surface of the catalyst support (Co^3+^ + e^−^ → Co^2+^). The O_2_ adsorbed on the catalyst surface dissociates into active oxygen species O_2_^2−^ and O^−^, which are further converted into lattice oxygen O^2−^. With the formation of lattice oxygen, Co^2+^ was oxidized to Co^3+^. In the Ru_4_/Cu1-Co1-O·MgO catalyst, the collaborative interaction between CuO as cocatalyst and Co_3_O_4_ further enhances the catalyst’s overall activity.

### 2.7. Separation and Purification of FDCA

Following the completion of the reaction, the filtrate obtained post-reaction was gathered, and the green separation of FDCA was achieved utilising semi-preparative liquid chromatography equipped with an UV detector. This method uses distinctions in interactions among different solutes as well as stationary and mobile phases to collect solutions containing FDCA selectively. A Nucifera C18 chromatography column (250 mm × 10 mm, 12 nm) was chosen, and the mobile phase consisted of methanol (30%) and 0.1 wt% formic acid water (70%). The column temperature was maintained at 30 °C. The semi-preparative liquid chromatogram of the separated FDCA was shown in Appendix A. A 60 mL reaction solution containing 0.3024 g of HMF should theoretically yield 0.3224 g of FDCA when the yield of FDCA is 86.1%. The solution containing FDCA was separated by semi-preparative liquid chromatography and underwent further processing with rotary evaporation to eliminate methanol, formic acid, and water. Subsequently, FDCA powder was obtained upon drying. The obtained actual weight of FDCA was 0.2417 g, yielding an actual yield of 75%. HPLC and ^1^H-NMR analysis reveals that the purity of FDCA exceeds 99% (^1^H-NMR as shown in Appendix A). Notably, this separation method is greener and more environmentally friendly than techniques reported in the literature, particularly those involving the addition of concentrated hydrochloric acid to adjust the pH for FDCA crystallisation.

## 3. Experimental Section

### 3.1. Preparation of the Catalyst

The composite metal oxide supports were obtained by dissolved 6 mmol of copper nitrate and cobalt nitrate in 60 mL of deionized water and adjusted molar ratios at 1:0, 1:1, 1:2, 2:1, and 0:1. While stirring, 1.5 mol/L of NaOH was incrementally added until the pH reached 10, at which point the addition was stopped. The resulting mixture in the autoclave underwent hydrothermal treatment at 180 °C for 4 h. Following the hydrothermal treatment, the mixture was cooled to room temperature naturally, then filtered and washed with deionized water until it achieved a neutral pH. The resultant precipitate was dried at 80 °C for 12 h, obtaining the hydroxide precursor powder designated as Cu_x_O_y_, Cu1-Co1-O, Cu1-Co2-O, Cu2-Co1-O, and Co_x_O_y_. Subsequently, 0.2 g of the hydroxide precursor powder was carefully ground with varied molar ratios of magnesium acetate (2, 4, 6, 8 mmol) and subjected to calcination at 400 °C for 4 h. This process resulted in the formation of the composite MgO support, designated as Cu-Co-O·MgO (2 mmol), Cu-Co-O·MgO (4 mmol), Cu-Co-O·MgO, and Cu-Co-O·MgO (8 mmol).

The impregnation–-reduction method was employed to prepare the Ru-loaded catalyst. A specific quantity of RuCl_3_·3H_2_O, corresponding to the mass fraction of Ru in the support at 1–5 wt%, was dissolved in 12.5 mL of deionized water. Subsequently, 0.5 g of the support was introduced into the solution. The mixture was immersed and dispersed for 12 h under conditions of an ice water bath and stirring. 1.5 mol/L NaBH_4_ was dissolved in a 0.5 wt% NaOH aqueous solution, in which the molar ratio of 20:1 of BH_4_ˉ in NaBH_4_ to Ru^3+^ in RuCl_3_·3H_2_O. The NaBH_4_ solution was then gradually added to the dispersion and stirred for 12 h to facilitate reduction. Following this, the mixture underwent filtration, was washed to achieve neutrality, and underwent vacuum drying for 12 h, obtaining the Ru-loaded catalyst, denoted as Ru_z_/Cu-Co-O·MgO (z = 1, 2, 3, 4, 5 wt%).

### 3.2. Catalytic Reaction Equipment and Product Detection

The oxidation of HMF was conducted in high-pressure autoclave, which was equipped with an internal temperature controller and pressure gauge. The specific quantity of the catalyst (0.02–0.1 g), deionized water (5–10 mL), and HMF (0.2–2 mmol) were added in the autoclave. Following the injection of oxygen (0–1 MPa), the mixture was subjected to magnetic stirring at 500 rpm, heated to a specified temperature (90–130 °C), and maintained for a certain time (0–16 h). The catalyst was separated after the reaction. The reaction solution was diluted with pure water, and the reactants were subjected to analysis used Shimadzu LC 20 AT HPLC, featuring a UV detector and Shimadzu Shim-pack VP-ODS C18 column (250 mm × 4.6 mm, 5 µm). The mobile phase consisted of methanol (30%) and 0.1 wt% formic acid water (70%), with column temperature set at 30 °C. Quantification was performed using the external standard method to calculate the conversion rate and yield.

## 4. Conclusions

In conclusion, the Ru_4_/Cu1-Co1-O·MgO catalyst was developed for the base-free catalytic oxidation of HMF to FDCA. The incorporation of MgO into the Cu1-Co1-O metal oxide increased the specific surface area but also the alkaline sites of the catalyst. Following the loading of active centres Ru on Cu1-Co1-O and Cu1-Co1-O·MgO, the catalyst experienced improvements in both the specific surface area and alkaline sites, and the catalyst exhibited better low-temperature reduction properties. Furthermore, the synergistic interaction between CuO and Co_3_O_4_ played a key role in enhancing the catalyst’s overall activity. Under specific conditions, namely, a copper-to-cobalt molar ratio of 1:1, a magnesium-compounding amount of 6 mmol, and 4 wt% Ru loading, the Ru_4_/Cu1-Co1-O·MgO catalyst demonstrated exceptional performance. Achieving a 100% conversion rate of HMF and an 86.1% yield of FDCA, the catalyst operated optimally under specified reaction conditions (12 h, 120 °C, O_2_ 1.0 MPa, 0.08 g of catalyst). Furthermore, after five cycles of experiments, it was observed that the Ru nanoparticles exhibited resistance to leaching and oxidation, and the dissolution of MgO was effectively suppressed. By utilising semi-preparative liquid chromatography, with distinctions in interactions between various solutes and the stationary and mobile phases, the solution containing FDCA was selectively collected. This process resulted in isolating FDCA powder with a purity exceeding 99%. Notably, this separation method is advantageous due to its environmentally friendly approach, in contrast to techniques reported in the literature that involve the addition of concentrated hydrochloric acid to adjust the pH for the crystallisation of FDCA.

## Figures and Tables

**Figure 1 molecules-29-03213-f001:**
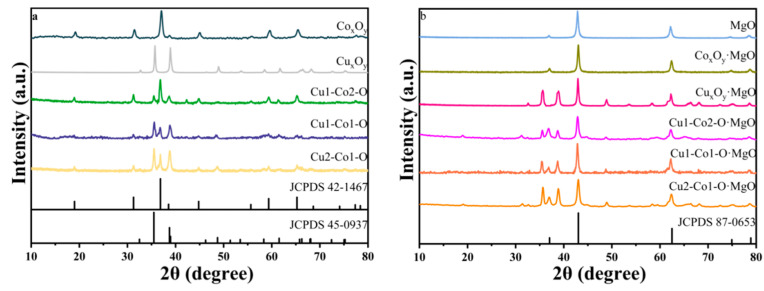
(**a**) XRD patterns of Co_x_O_y_, Cu_x_O_y,_ and Cu-Co-O. (**b**) XRD patterns of MgO, Co_x_O_y_·MgO, Cu_x_O_y_·MgO, and Cu-Co-O·MgO.

**Figure 2 molecules-29-03213-f002:**
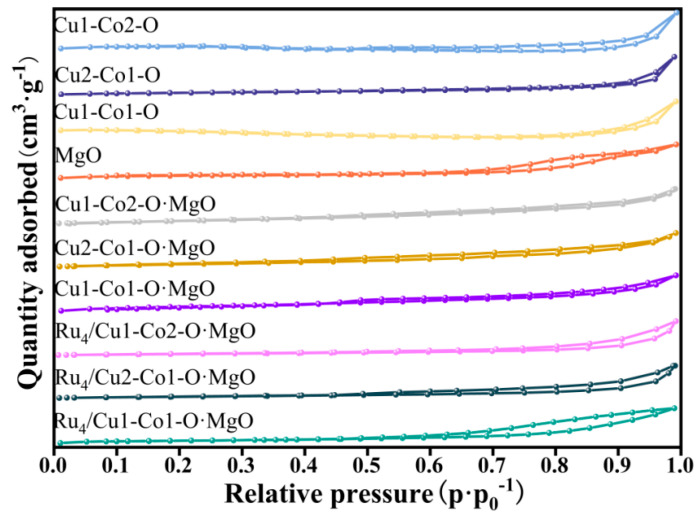
Nitrogen adsorption–desorption isotherms of different catalysts.

**Figure 3 molecules-29-03213-f003:**
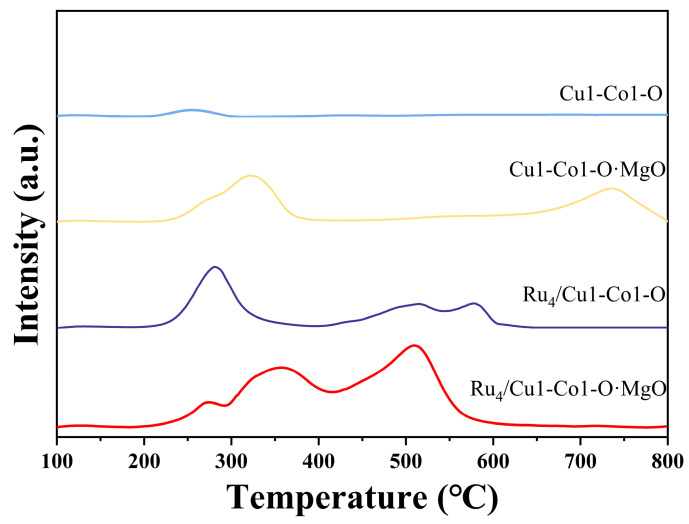
CO_2_-TPD spectra of different catalysts.

**Figure 4 molecules-29-03213-f004:**
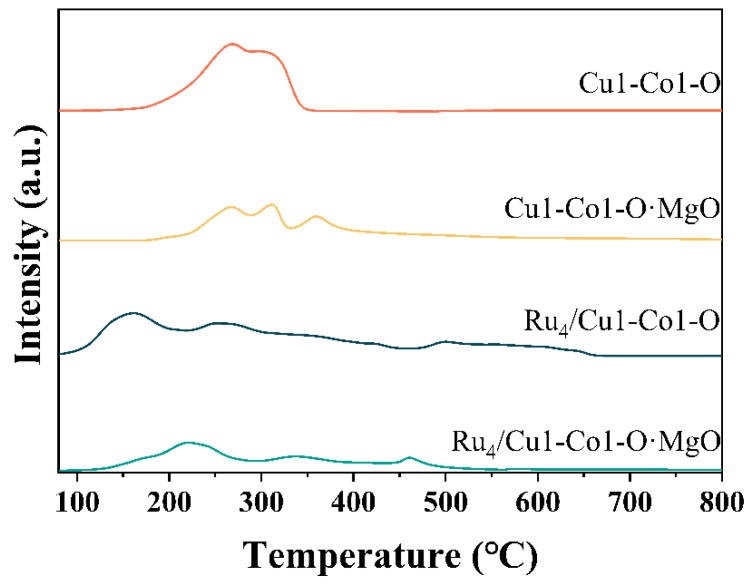
H_2_-TPR profiles for various catalysts.

**Figure 5 molecules-29-03213-f005:**
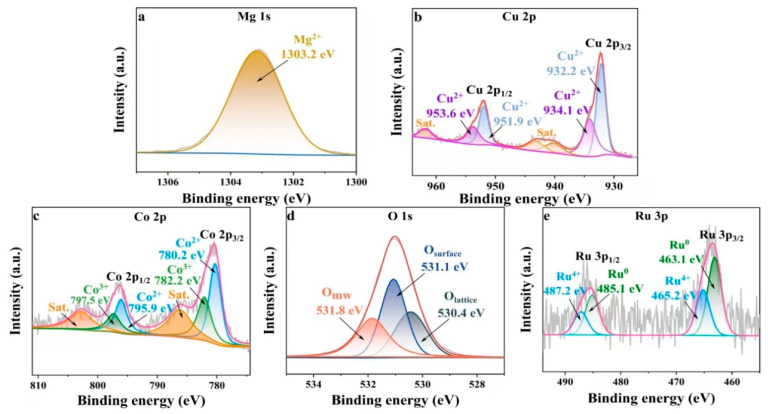
XPS spectra of Ru_4_/Cu1-Co1-O·MgO, featuring (**a**) Mg 1s, (**b**) Cu 2p, (**c**) Co 2p, (**d**) O 1s, and (**e**) Ru 3p.

**Figure 6 molecules-29-03213-f006:**
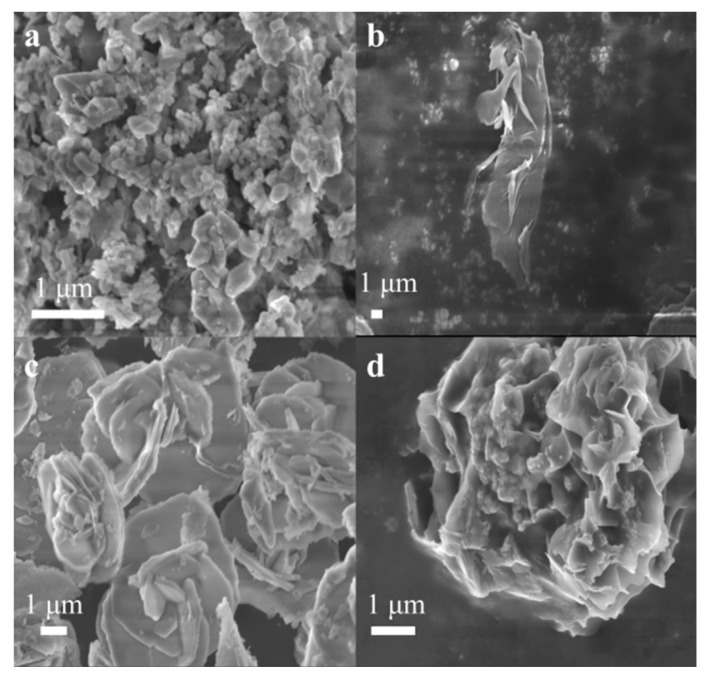
SEM images of (**a**) Cu1-Co1-O, (**b**) MgO, (**c**) Cu1-Co1-O·MgO, and (**d**) Ru_4_/Cu1-Co1-O·MgO.

**Figure 7 molecules-29-03213-f007:**
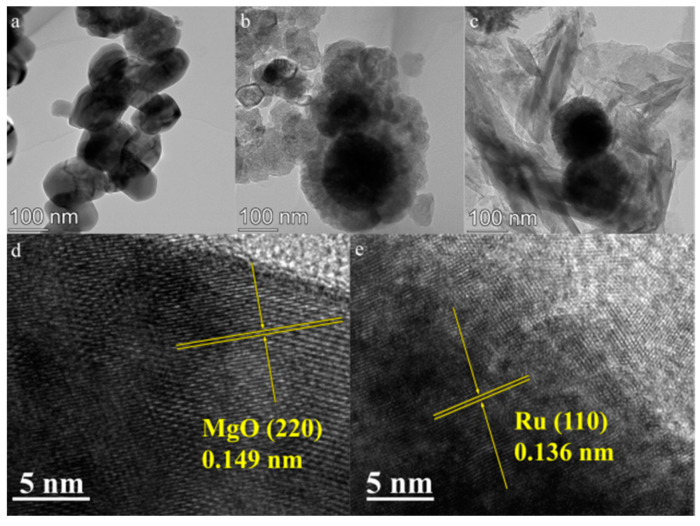
TEM images of (**a**) Cu1-Co1-O, (**b**,**d**) Cu1-Co1-O·MgO, and (**c**,**e**) Ru_4_/Cu1-Co1-O·MgO.

**Figure 8 molecules-29-03213-f008:**
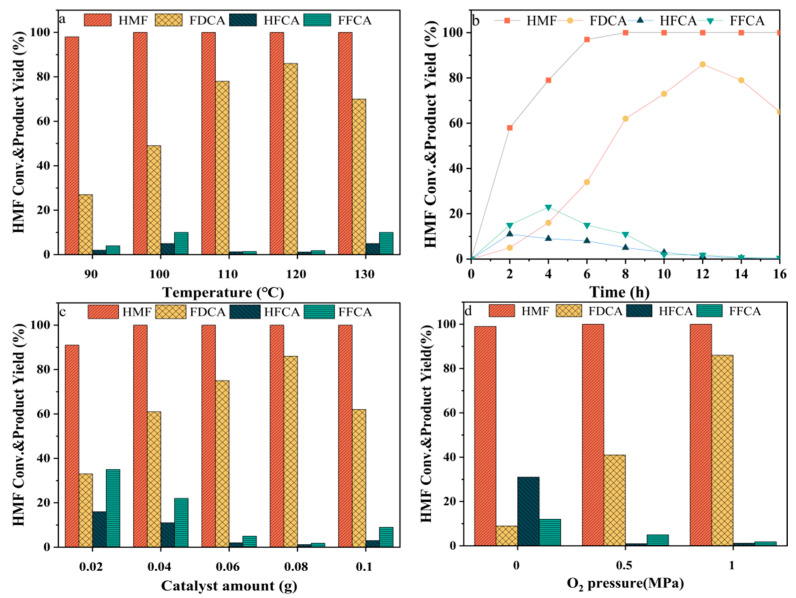
The influence of reaction conditions on the oxidative reaction of HMF. (**a**) The effect of reaction temperature under the following conditions: 0.08 g Ru_4_/Cu1-Co1-O·MgO, 1.0 MPa O_2_, and 12 h. (**b**) The effect of reaction time under the following conditions: 0.08 g Ru_4_/Cu1-Co1-O·MgO, 1.0 MPa O_2_, and 120 °C. (**c**) The effect of the amount of catalyst under the following conditions: 1.0 MPa O_2_, 120 °C, and 12 h. (**d**) The effect of oxygen pressure under the following conditions: 0.08 g Ru_4_/Cu1-Co1-O·MgO, 120 °C, and 12 h.

**Figure 9 molecules-29-03213-f009:**
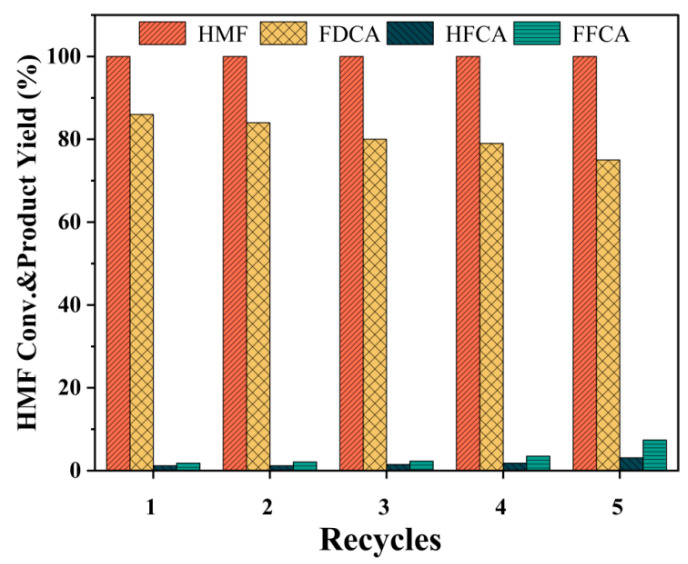
Cyclic stability of Ru_4_/Cu1-Co1-O·MgO under optimal conditions.

**Figure 10 molecules-29-03213-f010:**
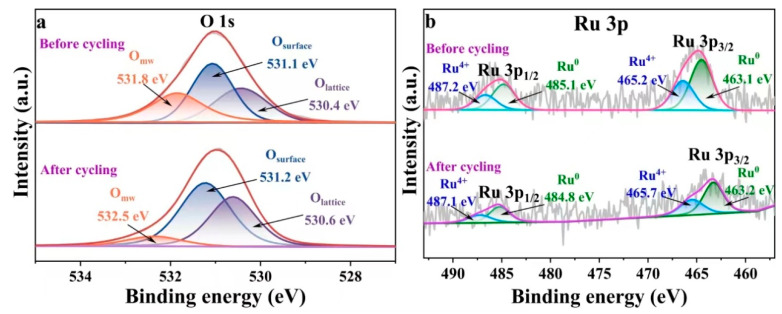
XPS spectra of Ru_4_/Cu1-Co1-O·MgO before and after cyclic experiments: (**a**) O 1s and (**b**) Ru 3p.

**Figure 11 molecules-29-03213-f011:**
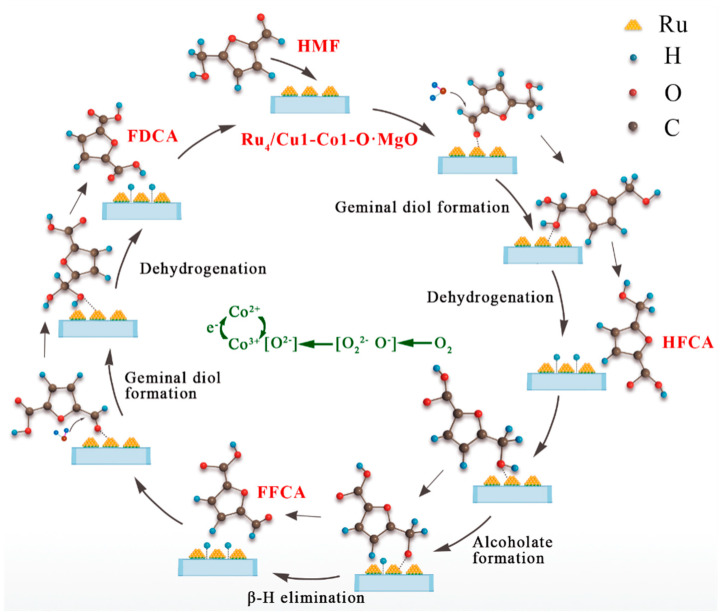
Possible reaction mechanism for the catalytic oxidation of HMF to FDCA facilitated by Ru_4_/Cu1-Co1-O·MgO.

**Table 1 molecules-29-03213-t001:** Specific surface area, pore volume, and average pore diameter of different catalysts and supports.

Sample	Surface Area (m^2^/g)	Pore Volume (cm^3^/g)	Average Pore Size (nm)
Cu1-Co2-O	1.3	0.0034	10.87
Cu1-Co1-O	4.0	0.0190	18.84
Cu2-Co1-O	6.6	0.0268	15.76
MgO	38.8	0.1544	16.72
Cu1-Co2-O·MgO	39.3	0.1264	2.73
Cu1-Co1-O·MgO	35.9	0.0677	7.49
Cu2-Co1-O·MgO	29.9	0.1070	3.83
Ru_4_/Cu1-Co2-O·MgO	44.4	0.2041	1.16
Ru_4_/Cu1-Co1-O·MgO	55.9	0.2193	15.68
Ru_4_/Cu2-Co1-O·MgO	63.4	0.3358	2.45

**Table 2 molecules-29-03213-t002:** CO_2_ consumption during the desorption of various catalysts.

Sample	Basicity (mmol/g)			
	Low Basicity (<250 °C)	Moderate Basicity (250–400 °C)	High Basicity (>400 °C)	Total
Cu1-Co1-O	0.001	0.001.	0	0.002
Cu1-Co1-O·MgO	0.002	0.055	0.058	0.115
Ru_4_/Cu1-Co1-O	0.005	0.043	0.040	0.088
Ru_4_/Cu1-Co1-O·MgO	0.009	0.132	0.152	0.293

**Table 3 molecules-29-03213-t003:** Screening of catalyst preparation conditions.

Entry	Sample	Conv. (%)	Y_FDCA_ (%)	Y_HFCA_ (%)	Y_FFCA_ (%)
1	Ru_4_/MgO	100	59.3	1.2	1.4
2	Ru_4_/Cu_x_O_y_·MgO	100	43.1	0.2	0.6
3	Ru_4_/Co_x_O_y_·MgO	100	18.6	1.1	1.5
4	Ru_4_/Cu1-Co2-O·MgO	100	22.4	1.2	1.8
5	Ru_4_/Cu2-Co1-O·MgO	100	33.7	0.5	1.3
6	Ru_4_/Cu1-Co1-O·MgO	100	78.6	1.3	1.4
7	Ru_4_/Cu1-Co1-O·MgO (2 mmol)	100	17.9	0.6	0.5
8	Ru_4_/Cu1-Co1-O·MgO (4 mmol)	100	42.0	0.3	0.6
9	Ru_4_/Cu1-Co1-O·MgO (8 mmol)	100	38.9	1.2	1.4
10	Ru_4_/Cu1-Co1-O	100	10.6	0.8	0.3
11	Cu1-Co1-O·MgO	72.7	3.5	2.2	25.7
12	Ru_1_/Cu1-Co1-O·MgO	84.8	30.2	15.3	22.6
13	Ru_2_/Cu1-Co1-O·MgO	100	35.4	1.7	2.5
14	Ru_3_/Cu1-Co1-O·MgO	100	71.0	4.3	2.7
15	Ru_5_/Cu1-Co1-O·MgO	100	63.7	9.1	2.0

Reaction conditions: 0.2 mmol HMF, 5 mL H_2_O, 0.08 g catalyst, 110 °C, 1.0 MPa O_2_, 12 h.

**Table 4 molecules-29-03213-t004:** Performance test of the catalyst for the oxidation of HMF at various temperatures.

Temperature (°C)	Conv. (%)	FDCA Yield (%)	TON ^1^	TOF ^2^ (h^−1^)
90	98.4	27.3	1.71	0.14
100	100	49.8	3.11	0.26
110	100	78.6	4.91	0.41
120	100	86.1	5.38	0.45
130	100	70.4	4.40	0.37

^1^: TON is determined by dividing the moles of FDCA produced by the moles of active sites in the catalyst. ^2^: TOF is then calculated by dividing the TON by the reaction time.

**Table 5 molecules-29-03213-t005:** Concentrations of Ru and Mg in the reaction solution.

Sample	[Mg^2+^] (mg/L)	[Ru^n+^] (mg/L)
Ru_4_/MgO	216.373	0.083
Ru_4_/Cu1-Co1-O·MgO	113.106	0.015

Reaction conditions: 0.08 g catalyst, 120 °C, 12 h, 1.0 MPa O_2_, 0.2 mmol HMF, 5 mL H_2_O.

## Data Availability

Data are contained within the article and Appendix A.

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
