# Peer review of "Base-Free Oxidation of HMF to FDCA over Ru/Cu-Co-O·MgO under Aqueous Conditions"

_molecules, 2024, doi:10.3390/molecules29133213_

Round 1
Reviewer 1 Report
Comments and Suggestions for Authors
The draft presented a catalytic systems of Ru based on base character supported catalyst MgO, together with Co-Cu-O synergetic effect. The prepared catalysts were applied in one of the most important reactions in biomass valorization. which is the oxidation of HMF into FDCA. The work is written and described well. After my revision, I recommended a minor revision: my comments as following:
How many mmoles of active sites on the surface of the catalyst (substrate/active site)?
How the authors make sure they don’t have over oxidation products CO and CO2? maybe carbon balance could help!
Did the authors used H2 for reduction step for Ru, may be its better than NaBH4 for the toxicity from my point of view?
In the tables which are presented the catalytic results the total yield less than 100% any details about the other products can improve the work?
Figure 2 and table 2 are overlapping in the original draft
Author Response
Response to Referee
Reviewer #1: Comments and Suggestions for Authors
The draft presented a catalytic systems of Ru based on base character supported
catalyst MgO, together with Co-Cu-O synergetic effect. The prepared catalysts were
applied in one of the most important reactions in biomass valorization. which is the
oxidation of HMF into FDCA. The work is written and described well. After my
revision, I recommended a minor revision: my comments as following:
Q1. How many moles of active sites on the surface of the catalyst (substrate/active
site)?
Answer: Thank you for your suggestion. Ru serves as the active center of the catalyst,
with the number of active sites corresponding to the molar amount of Ru. In the
catalytic oxidation of HMF to synthesize FDCA, 0.08 g of catalyst containing 4 wt%
Ru is required, resulting in a Ru mass of 0.0032 g. Using formula 1, the molar amount
of Ru is calculated to be 0.000032 moles. For further calculations of TON and TOF,
please refer to page 11.
The Moles of Ru can be calculated using the formula (1).
Moles of Ru = Mass of Ru/Molar mass of Ru (1)
Q2. How the authors make sure they don’t have over oxidation products CO and CO2?
maybe carbon balance could help!
Answer: Thank you for your suggestion. The oxidation of HMF to FDCA may not
lead to the formation of CO or CO2. The carbon balance not reaching 100% is due to
other side reactions occurring during the process, leading to the formation of products
other than HFCA, FFCA, and FDCA, such as formic acid and levulinic acid[1]. For
information on the impact of other side reactions on carbon balance, please refer to
page 11, in lines 359-361, 375-377 and 386-388.
[1]Ma, Y.; Zhang, T; Chen, L; Cheng, H.; Qi, Z. Self-developed fabrication of
manganese oxides microtubes with efficient catalytic performance for the selective
oxidation of 5-hydroxymethylfurfural[J]. Industrial & Engineering Chemistry
Research, 2019, 58(29): 13122-13132.
Q3. Did the authors used H2 for reduction step for Ru, may be its better than NaBH4
for the toxicity from my point of view?
Answer: Thank you for the kind suggestion. This is a very promising direction to
explore. We will incorporate this approach in our upcoming experiments to verify its
feasibility.
Q4. In the tables which are presented the catalytic results the total yield less than
100% any details about the other products can improve the work?
Answer: Thank you for your suggestion. The overall yield being below 100% might
be due to the formation of other by-products during the oxidation of HMF to FDCA.
There is literature evidence suggesting that changes in reaction conditions can lead to
the production of by-products, such as formic acid and levulinic acid, during the
oxidation of HMF to FDCA, thereby resulting in a decrease in the overall yield. The
impact of other by-products on the overall yield can be found on page 11, in lines
359-361, 375-377 and 386-388.
Q5. Figure 2 and table 2 are overlapping in the original draft
Answer: Thank you for your suggestion. We apologize for the inconvenience caused
by our formatting oversight, which resulted in a poor review experience. We have
revised the Figure 2 to ensure they no overlapping. The revised Figure 2 and table 2
can be found on page 6.

Reviewer 2 Report
Comments and Suggestions for Authors
The authors described a base-free oxidation of HMF to FDCA over Ru/Cu-Co-O·MgO under aqueous conditions. This study introduces a novel Ru-loaded Cu-Co-O·MgO catalyst achieving 100% HMF conversion and 86% FDCA yield under base-free conditions, enhancing catalyst stability and promoting green chemistry. The explanation of the results, characterization, and scientific novelty is very high. I recommend its publication after minor revision. Some suggestions and/or comments are included:
(1) See the Figure 2. It overlaps with the results presented in Table 2 of the PDF version.
(2) See the Table 4. The increase in FDCA yield at higher temperatures can be better explained. Are there theoretical calculations of activation energy for this oxidation reaction to support these results?
(3) See 2.7. Separation and purification of FDCA. The authors mention that the solution containing FDCA was separated by semi-preparative liquid chromatography, resulting in 75% yield and the purity of FDCA exceeds 99%. Some groups working in organic catalysis quantified yield, selectivity, and conversion using 1H NMR analysis. Given the high purity, I suggest including HRMS and 1H NMR spectra in CDCl3, DMSO-d6, and CD3OD (overlap from 0 to 12 ppm). This information would be valuable to other researchers, as it is not commonly reported. In my experience, obtaining such data in the past was challenging, often resulting in impure compounds. This addition could enhance the manuscript and attract more citation. It can be included in the Supporting Information.
(4) See the Supporting information. I suggest transferring Figure S1 and S2 to the manuscript.
(5) See the Figure 1. The size and resolution could be enhanced.
Author Response
Reviewer #2: Comments and Suggestions for Authors
The authors described a base-free oxidation of HMF to FDCA over Ru/Cu-Co-O·MgO under aqueous conditions. This study introduces a novel Ru-loaded Cu-Co-O·MgO catalyst achieving 100% HMF conversion and 86% FDCA yield under base-free conditions, enhancing catalyst stability and promoting green chemistry. The explanation of the results, characterization, and scientific novelty is very high. I recommend its publication after minor revision. Some suggestions and/or comments are included:
Q1. See the Figure 2. It overlaps with the results presented in Table 2 of the PDF version.
Answer: Thank you for your suggestion. We apologize for the inconvenience caused by our formatting oversight, which resulted in a poor review experience. We have revised the Figure 2 to ensure they no overlapping. The revised Figure 2 and table 2 can be found on page 6.
Q2. See the Table 4. The increase in FDCA yield at higher temperatures can be better explained. Are there theoretical calculations of activation energy for this oxidation reaction to support these results?
Answer: Thank you for your suggestions. As the reaction temperature increases, the reaction rate also increases, which is consistent with the literature reports [1]. However, higher reaction temperatures can lead to further reactions of FDCA, resulting in the formation of humins, which may be the reason for the decrease in TOF values at 130 ℃. Regarding the calculation of activation energy, we have not addressed this aspect yet. However, we will prioritize this research in our future work. Once again, we appreciate your suggestion. For information on the effect of reaction temperature on the catalytic reaction, please refer to page 11, in lines 359-361, 375-377 and 386-388.
[1]Gao, T.; Zhang, H.; Hu, C.; Jing, F.; Fang, W. Base-free aerobic oxidation of 5-hydroxymethylfurfural on a Ru (0) center in cooperation with a Co (II)/Co (III) redox pair over the one-pot synthesized Ru–Co composites. Ind. Eng. Chem. Res. J. 2020, 59: 17200-17209.
Q3. See 2.7. Separation and purification of FDCA. The authors mention that the solution containing FDCA was separated by semi-preparative liquid chromatography, resulting in 75% yield and the purity of FDCA exceeds 99%. Some groups working in organic catalysis quantified yield, selectivity, and conversion using1H NMR analysis. Given the high purity, I suggest including HRMS and 1H NMR spectra in CDCl3, DMSO-d6, and CD3OD (overlap from 0 to 12 ppm). This information would be valuable to other researchers, as it is not commonly reported. In my experience, obtaining such data in the past was challenging, often resulting in impure compounds. This addition could enhance the manuscript and attract more citation. It can be included in the Supporting Information.
Answer: Thank you for your sincere suggestions. We have added the 1H-NMR analysis in the Supporting Information, as shown in Figure S3.
Q4. See the Supporting information. I suggest transferring Figure S1 and S2 to the manuscript.
Answer: Thank you for your sincere suggestions. We have transferred Figure S1 and S2 from the supporting information into the manuscript. The Figure 2 (Figure S1) and Figure 3 (Figure S2) to be included in the manuscript can be found on page 5.
Q5. See the Figure 1. The size and resolution could be enhanced.
Answer: Thank you for your suggestion. I have increased the size and resolution of the Figure 1 as required. The revised Figure 1 can be found on page 3.
